# Brief communication: Tropical glaciers on Puncak Jaya (Irian Jaya/West Papua, Indonesia) close to extinction

David Ibel[1], Thomas Mölg[1], Christian Sommer[1]

[1]Institute of Geography, Friedrich-Alexander-Universität Erlangen-Nürnberg (FAU), 91058 Erlangen, Wetterkreuz 15, Germany

*Correspondence to*: David Ibel (david.ibel@fau.de)

**Abstract.** The majority of glaciers have been retreating for many decades on a global scale due to anthropogenic climate change, including the mostly small glaciers in the Tropics. In this brief report, we document area changes of the Puncak Jaya glaciers in South-East Asia on West Papua, Indonesia, until the present. The survey was based on recent high resolution multispectral satellite imagery of PlanetScope and Pléiades missions from 2023 and 2024. Additionally, we digitized and georeferenced historical glacier extents from analogue maps, resulting in a new overview map of glacier change on Puncak Jaya since 1850. The results show a decrease of total glacier surface area by more than 99 % since 1850 and by ~65 % since the last survey in 2018. In 2024, glacier area was 0.165 km$^2$ ± 5 %. The development of Puncak Jaya glaciers is thus in line with the global shrinkage of (tropical) glaciers. Assuming the current area retreat rates to continue, it is very likely that Puncak Jaya glaciers will disappear around 2030.

## 1 Introduction

Global air temperatures have risen due to anthropogenic climate change (World Meteorological Organization, 2022). While the global annual mean near-surface air temperature of 2022 climbed 1.15°C above the 1850-1900 pre-industrial average, corresponding to a global warming rate of 0.2 ± 0.1ºC per decade, surface air temperatures in mountainous areas show an enhanced increase of 0.3 ± 0.2ºC per decade (Hock et al., 2022; World Meteorological Organization, 2022). Moreover, atmospheric warming varies by region. In South-East Asia, mean annual air temperatures already started to rise between 1870 and 1940 and were 0.46°C above the 1961-1990 average in 2022 (Allison and Kruss, 1977; World Meteorological Organization, 2022). In Indonesia, where Puncak Jaya glaciers are located, mean daily maximum air temperatures increased by 0.18°C per decade between 1983 and 2012 (Supari et al., 2017). Regardless of region, glaciers around the globe have been in a mode of mass loss for many decades and tropical glaciers are no exception, as shown in recent surveys of tropical Andean and East African glaciers (Hock et al., 2022; Hinzmann et al., 2024; Fox-Kemper et al., 2021; Turpo Cayo et al., 2022). Some tropical glaciers even ceased to exist, e.g. the Conejeras Glacier, Colombia, which disappeared between 2023 and 2024 (World Glacier Monitoring Service WGMS, 2024).

Glaciers at low altitude are termed "tropical" when the following criteria are fulfilled (Kaser and Osmaston, 2002). First, they must be located in the astronomic tropics between the Tropic of Cancer and the Tropic of Capricorn (23°26'13.3'' N; 23°26'13.3'' S); second, they must be located within the oscillation range of the Intertropical Convergence Zone (ITCZ); third, the annual range of the air temperatures must be equal to, or smaller than, the daily temperature range. Currently, three areas fulfill these criteria. The Andes where the majority of tropical glaciers is located, East Africa and New Guinea (Kaser and Osmaston, 2002). On New Guinea, glacier ice only exists in the West Papua region, formerly known as Irian Jaya region, located in the West of the New Guinea Highlands on the rugged Sudirman range around Puncak Jaya peak (4.884 m a.s.l.) (Figure 1a) (Permana et al., 2019). These glaciers are located in one of Earth's wettest regions (2.500 – 4.500 mm yr$^{-1}$ precipitation), are strongly influenced by the El Niño Southern Oscillation and the South Pacific Convergence Zone and are the only remaining tropical glaciers in the West Pacific Warm Pool (Prentice and Hope, 2007; Permana et al., 2019).

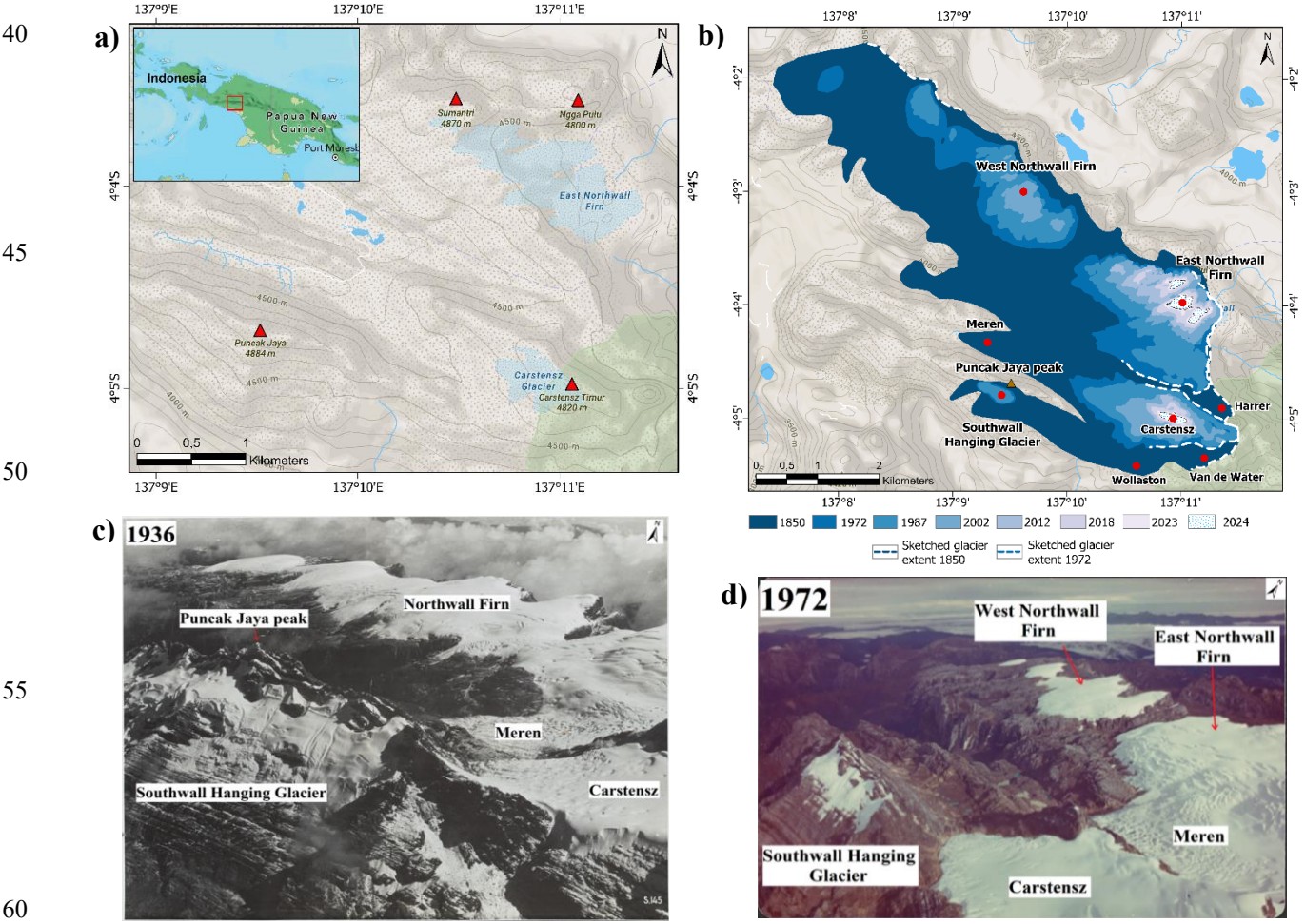

Figure 1: The geographic setting of glaciers around Puncak Jaya. **a):** Overview of glaciers and surrounding peaks (marked with red triangles) in the Puncak Jaya mountains. Puncak Jaya, with an altitude of 4.888 m a.s.l., represents

the highest peak in the region. East Northwall Firn glacier is located close to (Gunung) Sumantri and Ngga Pulu peaks, while Carstensz glacier lies next to Carstenz Timur peak. Glacier extents in the map are from 2005 (Openstreetmap Base map with contour lines (© OpenStreetMap contributors 2024. Distributed under the Open Data Commons Open Database License (ODbL) v1.0.), ESRI, NASA). b): Glacier changes between ~1850 to 2024, based on data of own surveys and of authors mentioned in Table S1. Uncertainties and estimates in the extents of 1850 and 1972, discussed by Peterson and Peterson (1994), are sketched with dashed lines (Openstreetmap Base map with contour lines (© OpenStreetMap contributors 2024. Distributed under the Open Data Commons Open Database License (ODbL) v1.0.), ESRI). c) and d): Oblique photographs, taken during the 1936 and 1972 expeditions, showing glacial retreat especially of the Northwall Firn, indicated by a bifurcation in 1972, separating the Northwall Firn into an East and West part (Dozy, J.J. (1936) and Carstensz Glacier Expedition (1972), Glacier Photograph Collection, National Snow and Ice Data Center, edited).

Tropical glaciers are interesting from a scientific point of view in several ways, as their changes are influenced by local climatic drivers as well as by prominent modes of macro- and mesoscale climatic dynamics (e.g. Mölg et al. 2020; 2009). Moreover, their high-altitude location, often above 4.000 m a.s.l. (Figure 1a) and thus close to the 600 and 500 hPa levels in the atmosphere, makes them good indicators of global climatic changes in the mid troposphere (Allison and Kruss, 1977; Mölg et al., 2009). Therefore, glacier systems located in the tropical Andes and in East Africa have been examined extensively in the last decades, focusing on glacier area changes and global and local climatic factors influencing glacial retreat. Recent examples are e.g. Carrivick et al. (2024), Gorin et al. (2024), Hinzmann et al. (2024), Mölg et al. (2020), Turpo Cayo et al. (2022). However, the glaciers on Puncak Jaya have received less attention as the only modern expeditions took place in 1973 and 2011 and the latest remote surveys were carried out between 2015 and 2018, creating a data gap to the present (Permana et al., 2019; Permana, 2011; Allison and Peterson, 1976). Thus, the present study aims to (i) close the existing data gap by examining the changes of glacier extent since the last surveys in 2015 and 2018 using high resolution multispectral satellite imagery, and (ii) construct a new map of glacial development from 1850 onwards by conducting a comprehensive data acquisition of (historical) analogue and digital data, which allows us to put the results of current glacier change into historical perspective.

**2 Brief history of Irian Jaya glacier studies**

During the Last Glacial Maximum (LGM) large parts of New Guinea's high mountain areas were covered by ice and snow (Kaser and Osmaston, 2002; Brown, 1990). The total area covered was about 2000 – 2200 km$^2$, including 863 km² in the Puncak Jaya area (Hope and Peterson, 1976). After several glacier advances and retreats during the Greenlandian and Northgrippian stage of the Holocene, glaciers in the Puncak Jaya mountains began to retreat by the end of the Little Ice Age (LIA) between 1850 and 1875 (Allison and Kruss, 1977; Bowler et al., 1976; Prentice et al., 2011). Interestingly, the

aforementioned glacier advances took place almost synchronously with neoglacial glacier advances on western Greenland and in the East African Rwenzori mountains (Bowler et al., 1976; Löffler, 1980; Peterson et al., 1973).

After the first report of glacier sightings in the Sudirman Range/Puncak Jaya mountains by the Dutch seafarer Jan Carstensz in 1623, the first expeditions to the glaciers did not take place until 1907 and 1909 (Wollaston, 1914; Temple, 1962; Hope, 1976). These expeditions were not able to reach the glaciers but documented their existence and extent through photography and cairns. After the first expedition that reached the glaciers in 1912, led by A.F.R. Wollaston and A. van de Water (Wollaston, 1914), the next expedition took place in 1936, led by A.H. Colijn and J.J. Dozy, which successfully climbed the summits of Carstensz Timur and Ngga Pulu (Coljin, 1937; Dozy, 1938). Thereafter, further ethnographic and geologic expeditions took place (Le Roux, 1948; Dozy et al., 1939), followed by several reconnaissance flights conducted by the US military during World War II (Allison, 1974; Ballard, 2001). During these expeditions, cairns indicating glacial extent were erected, and photographs were taken that were of great value for understanding glacier change on Puncak Jaya (Figure 1b,c,d).

The first ascent of Puncak Jaya mountain itself and the surrounding glaciers was accomplished by Heinrich Harrer in 1962 (Harrer, 1963; Hope, 1976). At this time, glacier retreat was already evident compared to the glacier extent in 1936 (Harrer, 1963). Between 1971 and 1973, a complete survey of all existing glacier areas on Puncak Jaya was conducted for the first time during two expeditions led by Australian researchers, finding the Southwall Hanging, Carstensz, Meren, East Northwall Firn and West Northwall Firn, Harrer, Wollaston and Van de Water glaciers (Figure 1b,c,d) (Allison and Peterson, 1976). The survey results of these expeditions indicated a further significant decline of glacier area from 13 km$^2$ in 1936 to 7.3 km$^2$ in 1972 and 6.4 km$^2$ in 1974 (Allison and Peterson, 1976).

From the 1970s onwards, satellite based remote sensing was increasingly used to measure glacier areas. SPOT satellite data, acquired in 1987, revealed a further reduction in total glacier area on Puncak Jaya to 5.09 km$^2$ (Klein and Kincaid, 2006; Peterson and Peterson, 1994). Between 1997 and 2000 Meren Glacier disappeared (Prentice and Glidden, 2010; Klein and Kincaid, 2006). The next survey, conducted in 2006 by using a time series of high-resolution IKONOS satellite data from 2000 to 2005, found a further reduction in total glacier area to 2.15 km$^2$ (Klein and Kincaid, 2006; Kincaid, 2007). During a survey in 2010, when ice core drilling was conducted on the glaciers to measure isotope composition, the disappearance of the last remnants of the Southwall Hanging Glacier was detected (Permana, 2011). The extents of the remaining glaciers East Northwall Firn and Carstensz were analysed again in 2018 using remote sensing imagery of PlanetScope mission, revealing a total surface area reduction from 0.653 km$^2$ in 2015, 0.546 km$^2$ in 2016 to 0.458 km$^2$ in 2018 (Permana et al., 2019).

Due to the remote location and difficult access of Puncak Jaya glaciers, only few in-situ mass change and ice thickness surveys have been conducted, mostly covering short time periods: In 1972, negative mass balances of –57 x 10$^3$ m$^3$ w.e. yr$^{-1}$ with a surface lowering of 0.064 m yr$^{-1}$ for the Carstensz glacier and –989 x 10$^3$ m$^3$ w.e. yr$^{-1}$ with a surface lowering of 0.509 m yr$^{-1}$ for the Meren/East Northwall Firn glacier were observed by the Carstensz Glacier Expedition (Allison, 1974). Maximum ice depth was estimated to be greater than 60 m for the East Northwall Firn Glacier, ~85 m for Meren Glacier and ~75 m for the Carstensz Glacier (Allison, 1974). Between 1995 and 1997, East Northwall Firn glacier experienced a negative

mass change of –4430 x $10^3$ m$^3$ w.e. yr$^{-1}$, whereas between 1997 and 2000 a slightly positive mass change of 452 x $10^3$ m$^3$
w.e. yr$^{-1}$ was surveyed (Prentice and Glidden, 2010). During an expedition in 2010, ablation stakes were placed on East
Northwall Firn Glacier, revealing an annual ice thickness loss of ~1.05 m (Permana, 2015). Ice thickness on East Northwall
Firn Glacier further decreased from ~30 m in 2010 to ~20 m in 2016, while an estimated ice thickness of ~6 m remained in
2022 (Permana et al., 2019; World Meteorological Organization, 2022). Although recent studies have not made statements
about glacier velocity due to lack of data, the continuous area loss of the glaciers indicates that they can be considered relict
ice.

## 3 Data selection and methods

As the last surveys of 2015 and 2018 showed small glacier areas of <0.2 km$^2$ at Puncak Jaya, our study employed high
resolution multi-spectral satellite imagery of PlanetScope mission (provided by Planet Labs PBC via the PlanetLabs
Education and Research Programme) with a horizontal resolution of 3.125 m and a near-daily revisit capacity (Planet Labs
PBC, 2023). Furthermore, very-high resolution multi-spectral satellite imagery of the Pléiades mission (provided by Airbus
DS through the European Space Agency) with a horizontal resolution of 0.5 m and a daily revisit capacity was used to
compensate for potential uncertainties (e.g. due to shading effects) in the lower resolution PlanetScope imagery (Airbus
Defence and Space Intelligence, 2021). Although no distinct accumulation and ablation seasons exist at Puncak Jaya glacier
area due to the homogenous inner-tropical climate setting, imagery selection focused on the time period from May to August
when rainfall and relative humidity minima can be expected (Permana, 2011). However, some images from outside this
season could also be considered (Table S2). Imagery selection focused on the absence of extensive cloud cover, shadowing,
or fresh snowfall, which limited the number of available image acquisitions due to the high degree of cloud cover in the
inner tropics and early morning cloud formation at Puncak Jaya (Prentice and Hope, 2007; Kaser and Osmaston, 2002). The
selected images (Table S2), which were already provided orthorectified, color-corrected and -optimized, were loaded into
Esri ArcGIS Pro (Version 3.1.3) software. A visual check of the orthorectification of the PlanetScope images was conducted
by comparing several distinct landmarks using Sentinel 2 imagery (Table S3). As deviations were detected (Forward Root
Mean Square Error of 0.76-3.60 m using 16 Control Points), georeferencing and image matching were applied using the
Sentinel 2 image to increase the survey's accuracy.

Due to the size of remaining glaciers on Puncak Jaya, we delineated the glacier outlines manually based on visual inspection
of all available high-resolution images using ArcGIS Pro, while using a scale range from 1:500 to 1:1200. Manual
delineation/digitization of glacier outlines, executed by an analyst using cursor tracking, is less influenced by disturbing
factors such as shading or debris cover, as it relies on the analyst's dynamic interpretation and visual verification. Despite the
fact that this approach is typically more time-consuming than an automated survey (e.g. using band arithmetic methods), it is
effective for small glacier areas with potential interferences, e.g. shading or snow cover, since visual validation is already
included in the process (Hinzmann et al., 2024). As we used high-resolution multispectral imagery and did not detect snow

cover, debris cover or shading issues, the ice areas were clearly distinguishable from surrounding rocky terrain and additional manual delineation runs were not necessary. However, to further increase the accuracy, we compared the results to very-high resolution Pléiades imagery and oblique photography from recent expeditions, whereby no deviations were detected. Previous works reported the range of uncertainty in glacier delineation to be between 2.3 % (Linsbauer et al., 2021), 3.3 % (Paul et al., 2020) and 5 % (Fountain et al., 2023). Hence, we assumed a maximum uncertainty of 5 % for our results, based on the findings of Paul et al. (2020; 2013). This approach was supported by a comparison between the results for total glacier area extent in 2018 by Permana et al. (0.458 ± 0.036 km²) and by our survey (0.468 ± 0.023 km²), resulting in a difference of ~2.2 % between the point estimates (Table S1) (Permana et al., 2019).

In addition to the acquisition of present-day glacier area using satellite imagery, a new digital map with historical glacier extents dating back to 1850 was generated. This map was primarily based on the one published by Peterson and Peterson (1994), which was an updated version of a map created during the Carstensz Glacier Expedition in 1972, compiling survey results of historic glacier extents back to the 19[th] century (Anderson, 1976; Hope et al., 1976). As Peterson and Peterson (1994) and Kincaid (2007) reported glacier mapping errors in the original 1972 map, which were also noticable in our study with deviations of up to hundreds of meters, both analogue maps from 1972 and 1994, after being scanned on a flatbed scanner, were georeferenced in ArcGIS Pro, using a Sentinel 2 image (Table S3) with 12 control points in Affine Transformation. These control points based on survey points created for a local plane coordinate system with arbitrary datum during the 1972 Carstensz Glacier Expedition (Anderson, 1976) and were transformed to WGS84 UTM 53S coordinates using Helmert transformation for our use. However, these 12 control points present on the updated map by Peterson and Peterson (1994) were unevenly distributed. Thus, the total RMSE was ~17 m, despite the addition of six control points based on geomorphological landmarks to account for the uneven distribution of control points. As the RMSE close to the glaciated areas was 3-7 m, and a visual inspection of a Sentinel 2 image (Table S3) to check for conformity with topographical characteristics supported the reliability of this result, the glacier areas for 1850, 1972 and 1987 were subsequently delineated manually and saved as polygon shapefiles. The glacier mapping errors in the 1972 map by Allison and Peterson (1976) left some data gaps in certain aspects of the mountain. These areas were sketched accordingly, following a combination of both maps by Allison and Peterson (1976) and Peterson and Peterson (1994) (Figure 1b).

Furthermore, glacier contours were manually delineated for the years 2012, 2016 and 2018 based on RapidEye and PlanetScope data (Table S4), as the contemporaneous outlines by Permana et al. (2019) were not available. Glacier extents of 2002 were provided by the study of Klein and Kincaid (2006).

## 4 Results and discussion

The trend of recession of Puncak Jaya glaciers since ~1850, documented in previous publications (Allison and Peterson, 1976; Harrer, 1963; Permana et al., 2019; Klein and Kincaid, 2006; Kincaid, 2007), has continued to the most recent years. The total glaciated area shrunk from 19.3 km$^2$ in 1850 to 6.4 km$^2$ in 1974 and 2.15 km$^2$ in 2002, and to 0.214 ± 0.011 km$^2$ in

2023 and 0.165 ± 0.008 km² in 2024 (Figure 2, Tables 1 and S1). Between 2018 and 2024 the total glacier area shrunk by 0.303 ± 0.015 km² or ~65 %. The western part of the East Northwall Firn shows the smallest glaciated area with only ~0.022 ± 0.001 km² in 2024 and is likely to disappear first in the years ahead (Figure 2).

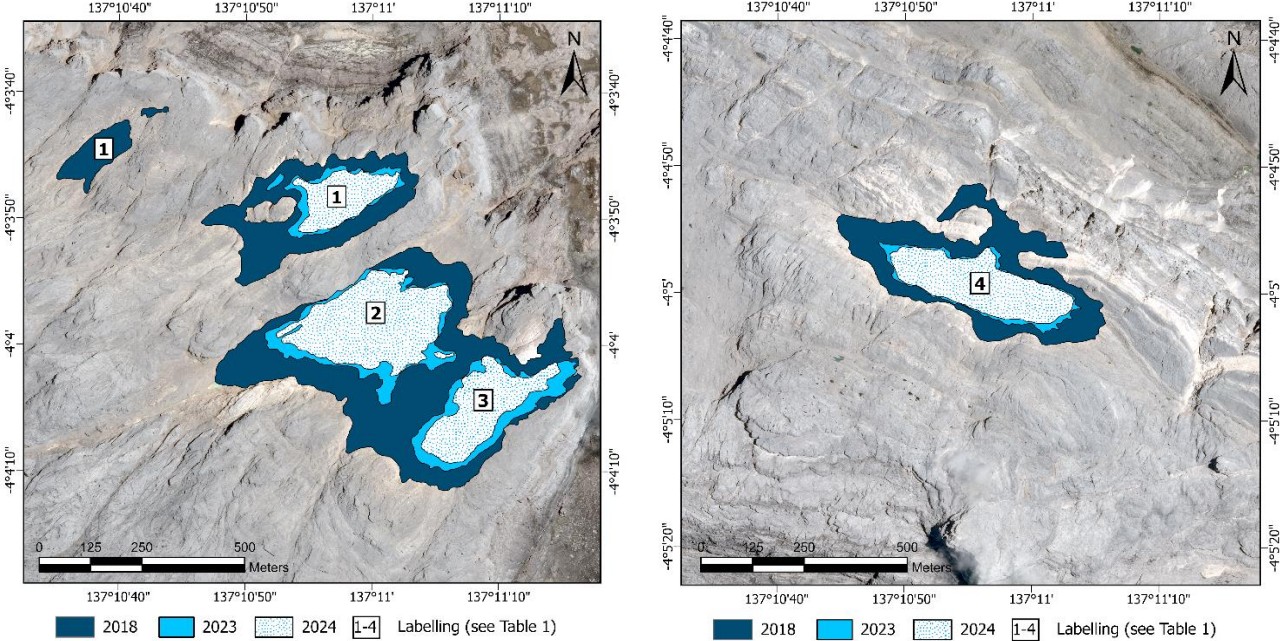

**Figure 2: Glacial retreat of East Northwall Firn glaciers (left) and Carstensz glacier (right) between 2018 and 2024. Labels 1-4 on glacier areas refer to Table 1 (Based on image data by © 2023, 2024 Planet Labs PBC and © CNES (2023), distribution Airbus DS, data provided by the European Space Agency. Background imagery of Pléiades mission provided by CNES (2023)). The original Pléiades image without annotations can be found in Figure S1.**

**Table 1: Overview of quantitative changes of Puncak Jaya glaciers. Note that East Northwall Firn Mid and East were one entity in 2018. "Label" refers to labelled glacier areas shown in Figure 2.**

| Label | Glacier | Area in m² | | |
|---|---|---|---|---|
| | | **2018** | **2023** | **2024** |
| 1,2,3 | East Northwall Firn | 367.977 | 163.413 | 120.812 |
| 1 | East Northwall Firn (West) | 96.228 | 29.473 | 21.707 |
| 2 | East Northwall Firn (Mid) | | 80.845 | 63.759 |
| 3 | East Northwall Firn (East) | 252.628 | 53.095 | 35.346 |
| 4 | Carstensz | 119.121 | 50.924 | 44.443 |
| **Total** | **Puncak Jaya** | **467.977 m²** | **214.337 m²** | **165.255 m²** |

| | | ~0.467 km² | ~0.214 km² | ~0.165 km² |
| --- | --- | --- | --- | --- |

Since the last survey in 2018, the westernmost part of the East Northwall Firn has disappeared and the main body of the East Northwall Firn has been separated into two parts by a bifurcation. All surveyed areas experienced area loss, while the East Northwall Firn experienced greater loss (0.247 km²) compared to the Carstensz glacier (0.074 km²) from 2018 to 2024 (Figure 2, Table 1), most likely due to the fact that the lost parts of the East Northwall Firn were located at a lower altitude than the Carstensz glacier area.

By putting the current survey's results into perspective to historical changes of Puncak Jaya glaciated areas, the drastic decline becomes evident (Figure 1b, Table S1). By 2002, the total glaciated area had declined by almost 90 % since 1850 (Klein and Kincaid, 2006) and in 2024 less than 1 % of the ~1850 glacier surface area remains. The former Northwall Firn Glacier split into two parts at some time between 1942 and 1962, resulting in the West Northwall Firn and East Northwall Firn part (Allison and Peterson, 1976). Meren Glacier vanished between 1997 and 2000 (Klein and Kincaid, 2006), while West Northwall Firn and Southwall Hanging glaciers disappeared between 2005 and 2012 (Figure 1b; Kincaid, 2007). Within the last two decades, the glaciers consisting of the remnants of East Northwall Firn and Carstensz glacier retreated to areas of the highest altitudes possible, close to the mountain tops of Carstensz Timur and Ngga Pulu.

The area loss rate shows a steady decline since ~1850 and indicates no climatological time period of glacier growth within the last 170 years (Figure 3). Despite a lack of data from the first half of the 21[st] century, a relatively steady retreat rate can be assumed for this period according to aerial photographies and cairn position surveys (Allison and Peterson, 1976). It can be noted that the loss rate slowed down between 2016 and 2024 compared to the recession between 2000 and 2015, as visible in Figure 3. However, this is typical of glaciers in a state of disappearance (Kaser et al., 2010). A comparison of the loss rate with the glacier area extent changes indicates that the observed slowdown between 2016 and 2024 happened after the complete loss of glacier parts at lower altitudes, with the remaining glacier parts at higher altitudes only, where more favourable climate conditions for glaciers exist.

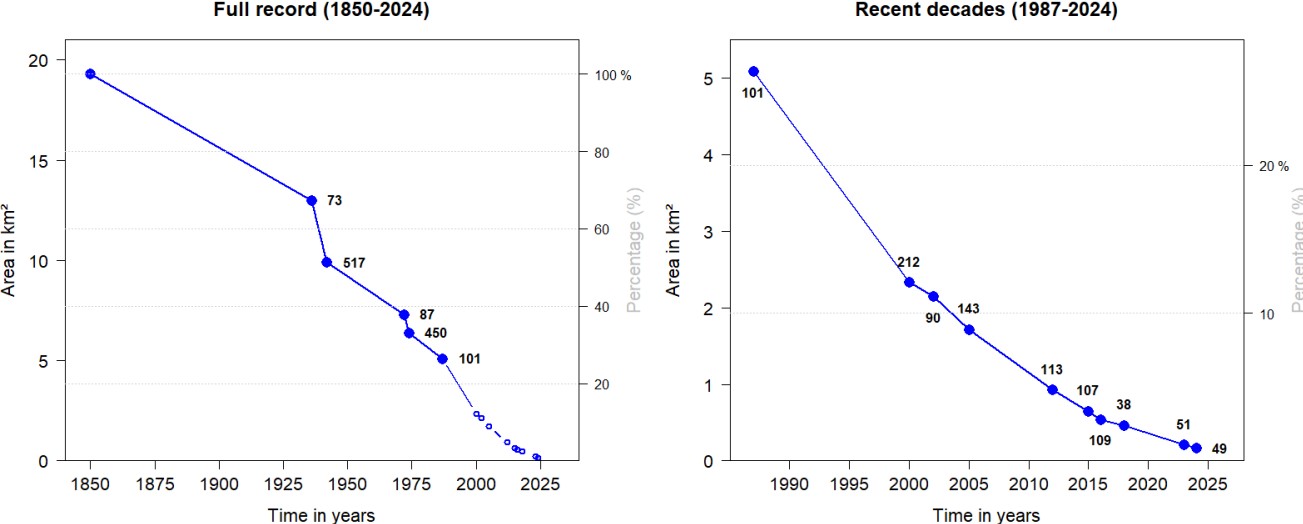

**Figure 3: Glacier area over time for all Puncak Jaya glaciers. The labels affixed to the data points correspond to the annual area loss rate ($10^3$ m² per year) in the previous time range. The left panel provides an overview over the full record from 1850 to 2024, while the right panel zooms into the most recent decades. Left y-axis represents total glacier area in km² and right y-axis represents the corresponding percentage.**

As important reason for the retreat of Puncak Jaya glaciers, the rise of mean annual air temperatures at the 550 hPa pressure level has been reported, with a rise of ~1°C alone at glacier altitude between 1972 and 2000, which has resulted in a near zero probability of average daily temperatures falling below freezing point on the glaciers after 1997 (Permana et al., 2019; Prentice and Hope, 2007). Additional factors, such as sea surface temperature changes, changes in the phase of precipitation due to air temperature rise, changes in radiation absorption as well as El Niño Southern Oscillation and monsoon variability, are playing further roles (Permana, 2015; Prentice and Hope, 2007; Permana et al., 2019; Kincaid, 2007). However, due to the lack of in-situ climatological datasets for the area, the direct local and regional climatic drivers are harder to identify (van Ufford and Sedgwick, 1998; Prentice and Hope, 2007). Based on the locations of the glacier remnants, the topography appears to have some influence, as all present ice masses are located on the western and southwestern ridges of their respective mountains, facing away from the morning sun (when skies are most likely cloud-free). This effect was already demonstrated by Hastenrath and Kruss (1988) to play a role for vanishing tropical glaciers.

## 5 Conclusion

In this brief communication, the strong recession of Puncak Jaya glaciers is documented, both for recent years and for the historic time range since the mid 19th century. New glacier surface areas for 2023 and 2024 on Puncak Jaya were determined,

which closes an existing data gap. Besides, we transferred historic glacier extent data into a digital format combined with a thorough analogue and digital data acquisition and collection, resulting in an up-to-date overview map of Puncak Jaya glacier history, which ranges from ~1850 to 2024. Using high- and highest resolution optical imagery for surveying glacier extent will become more important in the future since small glaciers, which will become more common due to climate

change, will be hard to survey using medium-resolution imagery (e.g. Hinzmann et al., 2024). It is expected that Puncak Jaya glaciers will disappear around 2030, if the observed annual area retreat rate since 2018 persists.

**Author contribution**

DI conducted the data analysis under supervision of CS and TM. The writing was led by DI, with contributions from TM and

CS. All authors discussed the results and edited the manuscript.

**Data availability statement**

The new glacier extents for 2023 and 2024 as well as the reanalysed earlier extents of 2012, 2016, 2018 and of the historic maps of 1972 and 1994 are openly available in PANGÄA (https://doi.pangaea.de/10.1594/PANGAEA.979847).

**Acknowledgements**

We thank Joni L. Kincaid for providing data of glacier extent in 2002 and Donaldi S. Permana (BMKG Indonesia) for providing recent photos of East Northwall Firn and Carstensz glaciers. We also want to thank Planet Labs PBC for providing access to PlanetScope imagery via their PlanetLabs Education and Research Programme and CNES/Airbus DS/European Space Agency for providing access to Pléiades imagery (Proposal ID: PP0093912). The constructive comments of two reviewers and the editor helped to improve the manuscript further.

**Competing interest**

At least one of the (co-)authors is a member of the editorial board of The Cryosphere.

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
