# Peer review of "Brief communication: Tropical glaciers on Puncak Jaya (Irian Jaya/West Papua, Indonesia) close to extinction"

_EGUsphere, 2025_

## Author Response (AR1)

*The Cryosphere*

**Brief communication: Tropical glaciers on Puncak Jaya (Irian Jaya/West Papua, Indonesia) close to extinction**

David Ibel*, Thomas Mölg, Christian Sommer

*corresponding author (E-Mail: david.ibel@fau.de)
* * *
Dear Editor,

thank you for inviting us to submit a revised version of our manuscript. For this response, we used our previous answers to the reviewer and editor comments (blue) and added the actual changes made in the revised manuscript (red). We also added a section containing enhancements and grammatical corrections in the revised manuscript. Changes made in our revised supplement are mentioned in the last section.

Thank you very much.

Sincerely,

David Ibel, Thomas Mölg, Christian Sommer
* * *
**I.    Referee #1 (Mauri Pelto (30 Mar 2025))**

**Specific comments**

7: Reword to be more accurate, recognizing that in many regions including New Zealand, Norway, Western North America and the European Alps glaciers did advance during portions 1950-1990s period. Hence it has not been many decades at a global scale. "Glaciers have been retreating for the last several decades on a global scale due to anthropogenic climate change, including the mostly small glaciers in the Tropics.

**Response:** We understand that a more specific sentence is needed while keeping it brief. Change can be made like "The majority of glaciers have been retreating for the last several decades on a global scale due to anthropogenic climate change, including the mostly small glaciers in the Tropics."

**Changes made:** We changed the sentence in line 7 to "The majority of glaciers have been retreating for many decades on a global scale due to anthropogenic climate change, including the mostly small glaciers in the Tropics."

9-12: Reword this is the abstract where sources/methods need not be reviewed. "The survey was based on recent 2023 and 2024 high resolution multispectral satellite imagery of PlanetScope and Pléiades missions, that were compared with digitized and georeferenced historical glacier extent, resulting in a new overview map of glacier change on Puncak Jaya since 1850."

**Response:** Sentence will be revised, shortened and specified: "The survey was based on recent 2023 and 2024 high resolution multispectral satellite imagery of PlanetScope and Pléiades missions, which were compared with historical glacier extents, resulting in a new overview map of glacier change on Puncak Jaya since 1850."

**Changes made:** Originally proposed sentence change for lines 9-12 was split into two sentences for enhanced readability: "The survey was based on recent high resolution multispectral satellite imagery of PlanetScope and Pléiades missions from 2023 and 2024. Additionally, we digitized and georeferenced historical glacier extents from analogue maps, resulting in a new overview map of glacier change on Puncak Jaya since 1850." DOI was removed from the Abstract section, as it is already present in the Data Availability Statement section.

17: The first paragraph discusses global temperature and regional temperatures. The focus of the paper is glacier change start your introduction with paragraph 2 and move the temperature portion down.

**Response:** We agree that this would be an option, but our preference would be to maintain the current structure. In our opinion, the current one has a higher logic flow (in cause/effect thinking).

**Changes made:** We kept our previous structure, as it appeared to provide a higher logic flow (in cause/effect thinking). One sentence in line 20 was slightly changed to enhance precision and an example of recent glacier extinction was added at the end of the paragraph.

Figure 2: Include a PlanetScope or Pléiades image, without the annotated coloring for the reader to be able to see the actual relict ice/glacier character. Given how cloudy the region is, such imagery would be a rare view for the community.

**Response:** We agree that providing such an image would be a valuable support for our paper and an interesting view for the reader. However, due to the limitations of figures for the "Brief Communication" format, we will provide this image in the Supplement.

**Changes made:** A Pléiades image, acquired in 2024, is provided in the Supplement (Figure S1).

170: Please move at least a condensed version of Table S1 into the main paper, leaving out years with limited observations. All the information in Table S5 can be put into this table as well. It is important to reference in the text here the area change of individual glaciers vs just the overall glacier area loss.

**Response:** Some important results of Table S1 are mentioned in the text. Due to figure and space limitation of our article type, it is not easy to move even a condensed version of Table S1 to the main text. Also, we believe that only providing the complete Table S1 gives the reader the option to fully resolve the glacial extent changes from 1850 to 2024. If there are no strong hesitations from the editor, we would prefer to keep Table S1 in full in the supplement, also because it is rather technical. However, we can consider to move Table S5 to the main text, which would be in the vein of the reviewer comment.

**Changes made:** We kept Table S1 in the supplement but moved Table S5 to now be Table 1 in the main text below Figure 2. We also enhanced readability of Table 1 by changing the table colouring.

199: Given the limited area and volume are any of the remaining four ice masses in Figure 2 still glaciers? Any evidence of movement from repeat imagery or crevassing? Either way please indicate that this may just be relict ice and not glaciers anymore. All ice will vanish by 2030, though the glaciers may well be gone now. The ice thickness is referenced in line 127, with your high resolution imagery, are there any indications or estimates of ice thickness?

**Response:** To the best of our knowledge, no movement was detected within the last decades but changes of crevasses haven't been checked in this study. We agree that it would be beneficial to add information about ice thickness changes. Due to the remote and rugged mountain area, however, expeditions for reading the ablation stakes have been rare and the results of such expeditions are presented in our preprint (127-129) to give the reader in insight into the ice thickness loss. Additionally, the coverage of this rugged terrain by satellites carrying radar or laser altimeters isn't good enough to conduct volume change measurements without a big effort (that would exceed the scope of our paper). We will, nonetheless, try to add a short sentence indicating the current absence of ice dynamics and consult relevant databases (e.g. Millan et al. 2022).

**Changes made:** We added the sentence "Although recent studies have not made statements about glacier velocity due to lack of data, the continuous area loss of the glaciers indicates that they can be considered relict ice." at the end of section 2, following the results of previous expeditions on ice thickness changes. Consulted databases (e.g. Millan et al. 2022) did not contain specific information about recent ice thickness changes of glaciated areas around Puncak Jaya.

218: For the remaining ice the orientation is indicated as being an important component is preservation. Is there any evidence of accumulation enhancement via wind drifting or avalanching.

**Response:** During our literature review we didn't find any hints on accumulation changes due to wind drifting or avalanches. However, due to the fast metamorphosis and densification of

fallen snow within hours or days due the tropical climate setting (with strong diurnal cycles), wind drift isn't expected to play an important role (which was found in a long, detailed project on the Kilimanjaro glaciers by one of the co-authors).

**Changes made:** No changes were made.

249: The paper begins appropriately discussing tropical glacier change. It is worth adding here other glaciers lost or nearly gone in this latitude belt. The World Glacier Monitoring Service reported that Conejeras Glacier, Colombia 4.8 N ceased to exist in 2024.

**Response:** We agree and will add examples of tropical glacier loss in the introduction.

**Changes made:** We added the example of the extinct Conejeras Glacier to our Introduction (Lines 27-28): "Some tropical glaciers even ceased to exist, e.g. the Conejeras Glacier, Colombia, which disappeared between 2023 and 2024."

**II.    Referee #2 (Anonymous (28 May 2025))**

**Major comments:**

Uncertainty analysis – While I can appreciate that this contribution is submitted as a 'brief communication', this should not prevent the authors from reporting on the uncertainties in their data and overall reports. Glacier mapping is not perfect, even when the quality of imagery is good, and the mapper is experienced. Glacier areas (and rates of change) are reported with no uncertainties. The uncertainty in the map from 1850 CE must also be high, but it sounded like only three 'landmarks' were used geocoding (authors mention distortion and shifts). What translation was used? Three ground control points for the map would yield an extremely high RMSE unless a thin plate spline was used (which assumes no errors, but this really would be unrealistic anyway). I would recommend that the authors consult papers that describe how to complete an uncertainty analysis for glacier mapping (e.g. Granshaw and Fountain, 2006; Fountain et al., 2023).

**Response:** We agree that it is important to assess the uncertainty of manually delineated surface areas. We suggest calculating the uncertainty based on the findings of Paul et al. (2013), who estimated the uncertainty in the outlines of glaciers in Alaska and the European Alps mapped from images with different spatial resolutions and by different analysts. For glaciers with a high contrast between the glacier surface and the surrounding ice, i.e. clean ice glaciers, as in the case of this study (see supplementary Figure S2), the authors reported an uncertainty in the digitised glacier area of ~5%. In line with this, we will calculate the uncertainty of each glacier and timestamp and include the respective values in the main text and the supplementary table S1.

We will conduct another uncertainty analysis for the 1850 CE map, but due to limited information regarding landmark positions a better result might not be possible.

**Changes made:** We calculated and added an uncertainty of 5 % to our surveyed areas, following the results of previous publications (e.g. Paul et al. 2013; 2020). We also compared our survey's results for total glacier area extent in 2018 with the results of Permana et al. (2019), revealing a difference of ~2.2 %, which supports our approach.

We also reanalysed our results for the 1850 glacier extents that were based on a map published by Anderson (1976). As glacier mapping errors in this map have been mentioned in two previous publications (Kincaid 2007; Peterson and Peterson 1994), we georeferenced an updated version of this map, published by Peterson and Peterson (1994), with a Sentinel 2 image (Table S3), using a combination of both maps and control points published by Anderson (1976), followed by visual check for conformity with topographical characteristics of the results. The map, shown in Figure 1b, was subsequently updated and enhanced by sketching areas with present data gaps as well as an updated colouring scheme. The main text (Lines 169-185) was updated as well, containing information about the RMSE and control points.

Attribution of loss – Why did the authors not attempt to attribute the observed area changes to climate drivers. The end of the discussion section points out a few possible factors which could explain these changes, but the reader is left wondering what the primary driver really is. They describe, for example, temperature changes between 1972-2000 but their data is mostly after 2000. It would be relatively easy to analyze temperature at the appropriate pressure level (they cite 550 hPa) from ERA5 over the period 1950-present for the closest grid point. Even a simple descriptive analysis may allow the authors to at least partition some of their results into climatic vs. non-climatic factors.

**Response:** While we fully agree that this would be an exciting question, in our opinion it exceeds the scope of a "brief communication" paper. It is true that there is experience in our author team as one of us is specialized in the attribution of cold-region climate change to atmospheric processes (T. Mölg). However, based on that previous work we argue that including a simple analysis would not cover the question sufficiently. We could include records from ERA5 for the location, but this would raise the issue and critique of reanalysis data not being representative (or only to a limited degree) for mountain regions. We feel that this would introduce further uncertainties, which are impossible to address within a "brief communication" (especially since we will have to include more on the uncertainty in glacier mapping, suggested by the same referee). Hence, the proposed separation of climatic versus non-climatic factors can, unfortunately, not be done within a simple analysis. See, for example, Mölg & Kaser (2011, https://doi.org/10.1029/2011JD015669) for more background on our argument.

**Changes made:** As argued in the response above, discussing the role of potential drivers of mass change would exceed the scope of a "brief communication" paper. Hence, no changes were made in our manuscript but we acknowledge that this is a knowledge gap that should be addressed in future works.

Short history section is too long – While the length of this section might be fine for a full-size manuscript, I found this section to be perhaps too rich in details (e.g. do we really need to know names of past explorers for the data presented in the paper?). I would prefer to see more details that pertain to supporting the objectives of the paper and/or the reliability of the results (uncertainties).

**Response:** We will try to shorten this section a bit. While working on this study, we noticed that bits and pieces of information about the glacier's history were scattered over different kinds of literature types and there was no good coverage of the history in one place. Hence we think that readers would appreciate this section.

**Changes made:** Sentences were restructured and compressed throughout section 2 to shorten it while preserving the content of historical information. Some information regarding glacier extents on other mountain ridges on Papua were deleted (Line 113).

**Minor points:**

Line 11: not certain what authors imply by 'accounts'

**Response:** We will replace accounts with extents.

**Changes made:** Replaced "accounts" with "extents".

Line 13: change to 'In 2024, glacier area was …'

**Response:** We will make this change

**Changes made:** Sentence changed to "In 2024, glacier area was $0.165 \text{ km}^2 \pm 5 \text{ %}$."

Line 14: Strike 'very' – a vague qualifier unless you are talking about a definition (e.g. 'very fine sand' is defined by a size range).

**Response:** We agree that qualifiers like "very" should be used with caution, but the term "very likely" is, in our opinion, accepted usage for the highest likelihood level following IPCC report terminology.

**Changes made:** We removed the word 'very' throughout the document except in line 14, as it has been deliberately chosen to indicate a high likelihood level of glacier area loss.

Line 27: Use 'First', 'Second,' … rather than 'Firstly' – less wordy.

**Response:** We will make this change.

**Changes made:** Wording has been changed according to the Reviewer's comment (Lines 29-32).

Line 73: Either refer to their position in terms of pressure level (e.g. 500-600 hPa) ,or actual elevation above sea level. Using both is confusing since they often don't coincide.

**Response:** We will add the actual elevation above sea level in this sentence.

**Changes made:** We added the actual elevation in meters above sea level to the main text, while referring to Figure 1a.

Line 80: As described in the major comments, I would recommend that the authors at least consult reanalysis data for attribution (at least annual or warm-season temperature anomalies).

**Response:** Please see our argument above.

**Changes made:** No changes were made.

Line 88: Do we know these glaciers actual surge? Or do the authors mean 'advance' or 'underwent expansion'?

**Response:** We will replace "surge" with "advances" as in the cited literature it is only explained that advances have happened but not whether these advances were surges.

**Changes made:** We replaced "surge" by "advances".

Line 111: Uppercase 'glacier' after Maren. Formal names use 'Glacier'. Same logic is Green Lake but Green and Blue lakes (lowercase when referring to plural).

**Response:** We will make this change.

**Changes made:** The word 'glacier' has been replaced with 'Glacier' throughout the manuscript wherever applicable.

Line 119: Strike 'very'

**Response:** We will replace "very" by "only".

**Changes made:** We replaced "very" by "only".

Line 120: I think the convention for units is to use superscripts 'no slash') so $m^{3}$ w.e. $yr^{-1}$ throughout

**Response:** We will make this change.

**Changes made:** We changed the unit convention to "$m^3$ w.e. $yr^{-1}$" throughout the whole manuscript.

Line 120 onward: The use of the term 'mass balance' when referring to mass (volume) change is confusing. If you don't have surface area I suggest you use 'mass change'.

**Response:** We will make this change.

**Changes made:** We exchanged "mass balance" by "mass change" where the used references didn't clearly state that mass balances were surveyed.

Line 140-145: There are few details provided so it become difficult for a reader to agree with your interpretation as to the quality of the co-registration. Can you provide some more information to support this claim (RMSE, number of control points used, etc)?

**Response:** We will provide the requested information.

**Changes made:** We performed georeferencing on all used satellite imagery and provided information accordingly, including Root Mean Square Error and the number of used control points.

Line 151: Strike 'very' and throughout paper.

**Response:** We will strike "very" throughout, except for "very likely" (see response above).

**Changes made:** We removed the word 'very' throughout the document except in line 14, as stated above.

Lines 145-160: As described in the major comments section, more information is required and some uncertainty estimates for the glacier mapping.

**Response:** Uncertainty estimates will be given according to the reply provided in the major comment section.

**Changes made:** Uncertainty estimates were added, as stated in our response to the major comment.

Lines 165: So if Permana et al., (2019) report on these glaciers, how do your extents compare to that work? This comparison would provide you with at least some assessment of the reliability of the mapping between these projects.

**Response:** Unfortunately, upon request this data could not be provided by Permana et al. (2019).

**Changes made:** The required data could not be provided by Permana et al. (2019), so a detailed comparison of the individual glacier extents was not possible. However, we compared our survey's result for total glaciated area in 2018 with the results of Permana et al. (2019), revealing a difference of ~2.2 % and indicating a good reliability of our result.

Line 190: Not a big deal, but deep blue typically used to demarcate water on maps. Can you simply use a given color with varied saturation levels ? Also, showing polylines is preferred rather than polygons as one can't see examine imagery or present-day ice.

Response: Due to the size of some glacier parts the use of polylines would make these parts almost invisible for the reader. Hence, we decided to use colour-filled polygons. We will consider changing the colour range.

Changes made: As stated previously, we kept using colour-filled polygons instead of polylines to keep even the smallest changes visible. Colour was changed to steel blue to maintain good visual differentiation between the polygons.

Line 208: You can't assert that there was no period of growth especially between 1850 CE and late 1940s since you have no data. Statement needs revision.

Response: Agree. Will be revised.

Changes made: We added the sentence "Despite a lack of data from the first half of the 21st century, a relatively steady retreat rate can be assumed for this period according to aerial photographies and cairn position surveys", as described by Allison and Peterson (1976).

Line 225: If you log right panel x axis, what year does area =0.0 km2? As described in major comments section, how has temperature (500 hPa pressure level) varied over this time? ERA5 goes back to the 1950s. You can download monthly data for seasonal, annual anomaly calculation.

Response: See reply to major comment above. We argue that using ERA5 without deeper analysis is not robust enough.

Changes made: We adjusted the right side x axis in both panels.

Line 235: There is a logic gap here. The authors suggest that attribution is not possible given the lack of meteorological data yet in the introduction there is mention of glaciers like this being good indicators of global climate. The two last authors have used reanalysis in the past. Why not do a simple analysis to help attribute changes of these glaciers?

Response: Again, as argued in the major comment reply, a simple analysis would not cover the question sufficiently and introduce further uncertainties. In the introduction, we refer to glaciers on a large-scale, for many of which this attribution was made. That is why in Line 235 we phrased the statement carefully for the Indonesian glaciers ("harder to identify").

Changes made: No changes were made.

Tables and Figures:

Fig 1 (a and b) are quite pixelated for me. Can they be reproduced at a higher resolution?

**Response:** Yes, will be done.

**Changes made:** Figure were added with higher resolution.

**III.  Author's response to Editor's remarks**

Addressing uncertainties in the analysis (R1: Ln 199; R2: Comment 1):

**Response:** We added an uncertainty of 5 %, based on the findings of Paul et al. (2020; 2013), to our survey's results in the main text and Supplementary Table S1. Moreover, we compared our survey's result for total glaciated area in 2018 with the results of Permana et al. (2019), revealing a difference of ~2.2 %.

We also reanalysed our results for the 1850 glacier extents, based on a map published by Anderson (1976). As glacier mapping errors in this map have been mentioned in two previous publications (Kincaid 2007; Peterson and Peterson 1994), we georeferenced an updated version of this map, published by Peterson and Peterson (1994), with a Sentinel 2 image (Table S3), using a combination of both maps and control points published by Anderson (1976), followed by visual check for conformity with topographical characteristics of the results. The map, shown in Figure 1b, was subsequently updated and enhanced by sketching areas with present data gaps as well as an updated colouring scheme.

Further development of the role of potential drivers of mass loss (R1: Ln 218; R2: Comment 2):

**Response:** Further discussing the role of potential drivers of mass change would, in our opinion, exceed the scope of a "brief communication" paper. A simple analysis using records from ERA5 would not cover the question sufficiently, introduce new uncertainties and raise the issue and critique of reanalysis data not being representative (or only to a limited degree) for mountain regions. Hence, we acknowledge that this is a knowledge gap that should be addressed in future works.

**IV. Further enhancements**

Line 36 – 38: Added some general geographical and climatological information on the surveyed area

Line 40: Map shown in Figure 1b) updated after georeferencing source maps from 1972 and 1994; glacier area extent of 1987 was added from Peterson and Peterson (1994); data gaps of glacier area extents in 1850 and 1972 were sketched with dashed lines

Line 67: Added information about data gaps in Figure 1b)

Line 91: Restructured sentence and added information (863 km² glaciated area)

Line 101: Added the sentence "which successfully climbed the summits of Carstensz Timur and Ngga Pulu"

Line 102: Restructured sentence and specified information about expedition types

Line 109-111: Restructured sentence and added the names of Harrer, Wollaston and Van de Water Glaciers

Line 113: Striked sentence "Additionally, during these expeditions several other small glaciers on Sudirman Range, namely on Mount Wilhelmina, Mount Juliana and Mount Idenburg, could be identified and surveyed from the air."

Line 114-112: Restructured sentences to enhance readability

Line 122: Replaced "0.552" with corrected result of "0.546"

Line 150-153: Added information about georeferencing results, including RMSE

Line 155: Added information about the scale range used for manual delineation

Line 192-194: Added uncertainty values and reference to Table 1

Line 200: Added the sentence "The original Pléiades image without annotations can be found in Figure S1."

Line 203: Corrected glacier extent values

Line 219: Added sentence "Despite a lack of data from the first half of the 21st century, a relatively steady retreat rate can be assumed for this period according to aerial photographies and cairn position surveys"

Line 226: Edited titles of Figure 3; added label and adjusted percentage numbers on right side x axis

Line 227-230: Exchanged "Annual area loss rate (103m2/a) of all" with "Glacier area over time for all"; added "The left panel provides an overview over the full record from 1850 to 2024" and "Left y-axis represents total glacier area in km² and right y-axis represents the corresponding percentage."

Line 257-258: Changed sentence to "The new glacier extents for 2023 and 2024 as well as the reanalysed earlier extents of 2012, 2016, 2018 and of the historic maps of 1972 and 1994"

Line 263-264: Added "The constructive comments of two reviewers and the editor helped to improve the manuscript further."

**V.    Grammatical Corrections**

Line 96: Corrected "east" to "East"

Line 97: Exchanged "description" with "report"

Line 98: Added "to the glaciers"

Line 100: Replaced "land markings" with "cairns" as this term was frequently used in historical literature

Line 100: Removed "successfully"

Line 104-105: Replaced "terrain markings" with "cairns" and "images" with "photographs"; striked "later on"

Line 127: Added "greater than"

Line 137: Replaced "extents" with "areas"

Line 145: Striked "without cloud cover"

Line 221: Changed "this rate" to "the loss rate"

Line 238: Added "in-situ"; added "and"

**VI.   Supplement changes**

Table S1: Added uncertainty values to our survey's results for 2012, 2016, 2018, 2023 and 2024

Table S2: Corrected information on the used PlanetScope 2023 scene

Table S5: Moved to now be Table 1 in the main text below Figure 2, as mentioned in the response above.

Image S1: Added to the supplement, as mentioned in the response above.

References:

Allison, I. and Peterson, J. A.: Ice areas on Mt. Jaya: Their extent and recent history, in: The equatorial glaciers of New Guinea: Results of the 1971 - 1973 Australian Univ., expeditions to Irian Jaya, survey, glaciology, meteorology, biology and palaeoenvironments, edited by: Hope, G. S., Peterson, J. A., Radok, U., and Allison, I., Balkema, Rotterdam, 27–38, 1976.

Anderson, E. G.: Topographic survey and cartography, in: The equatorial glaciers of New Guinea: Results of the 1971 - 1973 Australian Univ., expeditions to Irian Jaya, survey, glaciology, meteorology, biology and palaeoenvironments, edited by: Hope, G. S., Peterson, J. A., Radok, U., and Allison, I., Balkema, Rotterdam, 15–26, 1976.

Kincaid, J. L.: An assessment of regional climate trends and changes to the Mt. Jaya glaciers of Irian Jaya, Master Thesis, Texas A&M University, 2007.

Millan, R., Mouginot, J., Rabatel, A., and Morlighem, M.: Ice velocity and thickness of the world's glaciers, Nat. Geosci., 15, 124–129, https://doi.org/10.1038/s41561-021-00885-z, 2022.

Paul, F., Barrand, N. E., Baumann, S., Berthier, E., Bolch, T., Casey, K., Frey, H., Joshi, S. P., Konovalov, V., Le Bris, R., Mölg, N., Nosenko, G., Nuth, C., Pope, A., Racoviteanu, A., Rastner, P., Raup, B., Scharrer, K., Steffen, S., and Winsvold, S.: On the accuracy of glacier outlines derived from remote-sensing data, Ann. Glaciol., 54, 171–182, https://doi.org/10.3189/2013AoG63A296, 2013.

Paul, F., Rastner, P., Azzoni, R. S., Diolaiuti, G., Fugazza, D., Le Bris, R., Nemec, J., Rabatel, A., Ramusovic, M., Schwaizer, G., and Smiraglia, C.: Glacier shrinkage in the Alps continues unabated as revealed by a new glacier inventory from Sentinel-2, Earth Syst. Sci. Data, 12, 1805–1821, https://doi.org/10.5194/essd-12-1805-2020, 2020.

Permana, D. S., Thompson, L. G., Mosley-Thompson, E., Davis, M. E., Lin, P.-N., Nicolas, J. P., Bolzan, J. F., Bird, B. W., Mikhalenko, V. N., Gabrielli, P., Zagorodnov, V., Mountain, K. R., Schotterer, U., Hanggoro, W., Habibie, M. N., Kaize, Y., Gunawan, D., Setyadi, G., Susanto, R. D., Fernández, A., and Mark, B. G.: Disappearance of the last tropical glaciers in the Western Pacific Warm Pool (Papua, Indonesia) appears imminent, Proceedings of the National Academy of Sciences of the United States of America, 116, 26382–26388, https://doi.org/10.1073/pnas.1822037116, 2019.

Peterson, J. A. and Peterson, L. F.: Ice retreat from the Neoglacial maxima in the Puncak Jayakesuma area, Republic of Indonesia, Zeitschrift für Gletscherkunde und Glazialgeologie, 30, 1–9, 1994.

---

## Author Response (AR2)

*The Cryosphere*

**Brief communication: Tropical glaciers on Puncak Jaya (Irian Jaya/West Papua, Indonesia) close to extinction**

David Ibel*, Thomas Mölg, Christian Sommer

*corresponding author (E-Mail: david.ibel@fau.de)
* * *
Dear Editor,

thank you for inviting us to submit a revised version of our manuscript. For this response, we added the actual changes made in the revised manuscript (red). Changes made in our revised supplement are mentioned in the last section.

Thank you very much.

Sincerely,

David Ibel, Thomas Mölg, Christian Sommer
* * *
**I.     Author's response to Editor's remarks**

Give the comments of Reviewer 2, please consider presenting the uncertainties in Table 1.

**Response:** We added uncertainties to Table 1 (Line 201).

Likewise, following the comments of Reviewer 2, please consider adding a sentence in the conclusion about the uncertainties in attribution. While the current manuscripts discuss possible causes and support these with citations, part of the value of a brief communication is to encourage additional work on a subject. Thus, I believe explicitly bringing to light this knowledge gap in the conclusion would increase the value of the manuscript.

**Response:** We added a sentence in the conclusion: "A detailed attribution of this glacier recession to climatological causes beyond the more general factors (outlined in Sect. 4) will, however, require additional studies (and in-situ data acquisition) due to the well-known scale problem for mountain regions (Mölg and Kaser, 2011)." (Line 250).

Ln 249-> I am a bit surprised by the fact that average daily temperatures are rarely below zero. While this statement is cited, can you confirm this? This comment is likely a reflection of my surprise, rather than an actual scientific matter.

**Response:** We added two sentences to Section 4, containing more information about this statement (Line 234-237). The information about near-zero probability of daily average temperatures dropping below freezing point at ~4400m a.s.l., cited from Permana et al. (2019), is in line with other works on tropical glaciers, e.g. by Nicholson et al. (2013), who found daily average temperatures to fall slightly below 0°C at ~ 4900 m a.s.l.

Ln 254-> Consider moving the sentence starting with "However" to the end of the paragraph. This may highlight the large point of this paragraph.

**Response:** We moved this sentence to the end of the paragraph.

**II.   Further enhancements**

Line 64: Corrected "glacier" to "Glacier"

Line 82: Replaced "modern" with "extensive"

Line 152: Added "RMSE" acronym

Line 201: Changed colouring in Table 1 to grayscale, as requested by Copernicus Editorial Office; Corrected transmission errors in cell East Northwall Firn/2018 and in the last row

Line 207: Corrected transmission error, changed "0.247" to "0.228"

Line 252-254: Enhanced wording of sentence to "Considering the increasing prevalence of small (tropical) glaciers due to climate change, high- and highest-resolution optical imagery will become more important for surveying glacier extent in the future in comparison to medium-resolution imagery"

Changed thousands separator to "," throughout the whole manuscript.

**III.  Supplement changes**

Table S1: Changed "glacier" to "Glacier" throughout the first column

Image S1: Changed title from "Supplementary Image S1" to "Supplementary Figure S1", as requested by Copernicus Editorial Office.

References:

Nicholson, L. I., Prinz, R., Mölg, T., and Kaser, G.: Micrometeorological conditions and surface mass and energy fluxes on Lewis Glacier, Mt Kenya, in relation to other tropical glaciers, The Cryosphere, 7, 1205–1225, https://doi.org/10.5194/tc-7-1205-2013, 2013.

Permana, D. S., Thompson, L. G., Mosley-Thompson, E., Davis, M. E., Lin, P.-N., Nicolas, J. P., Bolzan, J. F., Bird, B. W., Mikhalenko, V. N., Gabrielli, P., Zagorodnov, V., Mountain, K. R., Schotterer, U., Hanggoro, W., Habibie, M. N., Kaize, Y., Gunawan, D., Setyadi, G., Susanto, R. D., Fernández, A., and Mark, B. G.: Disappearance of the last tropical glaciers in the Western Pacific Warm Pool (Papua, Indonesia) appears imminent, Proceedings of the National Academy of Sciences of the United States of America, 116, 26382–26388, https://doi.org/10.1073/pnas.1822037116, 2019.